# Long-Term Improvement of Gait Kinematics in Young Children with Cerebral Palsy Treated with Botulinum Toxin Injections and Integrated/Intensive Rehabilitation: A 5-Year Retrospective Observational Study

**DOI:** 10.3390/toxins17030142

**Published:** 2025-03-15

**Authors:** Weronika Pyrzanowska, Magdalena Chrościńska-Krawczyk, Marcin Bonikowski

**Affiliations:** 1Health Sciences Department, The Maria Grzegorzewska University, 02-353 Warsaw, Poland; wpyrzanowska@aps.edu.pl; 2Department of Children’s Neurology, University Children’s Hospital, 20-093 Lublin, Poland; magdalenachroscinskakrawczyk@umlub.pl

**Keywords:** cerebral palsy, integrated rehabilitation, botulinum toxin-A, OGS, gait kinematics

## Abstract

Introduction: Patients with cerebral palsy (CP) present mobility limitations that alter their activity and participation in social life. This study aimed to assess changes in gait kinematic measurements using the Observational Gait Scale (OGS) and preselected parameters related to the foot, knee, and hip in children with CP who received repeated BoNT-A injections within a rehabilitation treatment over a five-year follow-up period. Material and methods: This single-center retrospective observational study included 200 consecutive children with bilateral CP (GMFCS I–IV). The five-year follow-up period was analyzed. Patients received between 5 and 10 BoNT-A treatments (mean 7.6 ± 2.3), with total doses per session varying from 20 units/kg to 30 units/kg for ABOBoNT-A and from 10 units/kg to 20 units/kg for OnaBoNT-A. In most cases, multilevel injections were performed, including in the hip flexors and adductors, knee flexors, and foot plantar flexors. Results: The mean age of the patients at the beginning was 32.23 months (±6.96). The OGS score improved in 74.5% and 76.5% of the patients, and deterioration occurred in 8.5% and 7% of patients for the right and left lower extremities, respectively. The changes in the OGS typically ranged from 1 to 4 points. Significant improvements in the knee position at midstance, initial foot contact, foot contact at midstance, timing of heel rise, and knee and hip positions at terminal stance were observed. Conclusions: The data from our retrospective observational study show a significant, long-term, positive effect of integrated treatment on gait kinematics in patients with CP in a homogenous group of young children with bilateral spastic cerebral palsy.

## 1. Introduction

Cerebral palsy (CP) describes “a group of permanent disorders of the development of movement and posture causing activity limitations, which are attributed to nonprogressive disturbances in the developing fetal or infant brain” [1]. Between 75 and 91% of patients with cerebral palsy (CP) have spasticity that interferes with motor function [2,3]. In CP, spasticity often coexists with weakness, affecting lower limb muscle groups and causing abnormal gait patterns, which could lead to secondary musculoskeletal deformities, increasing the patient’s functional impairment [3]. Over the past thirty years, botulinum toxin type A (BoNT-A) has become a significant treatment option for hypertonia in children with cerebral palsy (CP), helping achieve maximal potential development and preventing secondary complications [4]. A major advancement in treating children with CP is the use of multi-level botulinum toxin type A (BoNT-A) injections, integrated into a comprehensive rehabilitation strategy that includes physiotherapy and orthotic interventions alongside various other treatments [3,5,6]. Such an integrated approach brings measurable results and changes the natural course of the disease [7,8]. The question of the long-term impact on gait is still valid [4]. This study aimed to assess changes in gait kinematic measurements using the Observational Gait Scale (OGS) and preselected parameters related to the foot, knee, and hip. The OGS was developed from the Physician’s Rating Scale (PRS) as a straightforward yet comprehensive tool for examining spastic gait. The primary purpose of the OGS is to evaluate gait parameters derived from video recordings, utilizing a structured scale. The OGS is particularly beneficial when children are either too small or insufficiently cooperative for instrumented gait analysis. Furthermore, this scale has proven highly effective when analyzing a typical walk observed on split-screen video in slow motion [9].

## 2. Results

The study group comprised 120 males and 80 females aged 24 to 46 months. The mean age was 32.23 months (±6.96). Before the treatment, 6 (3%) children were rated at GMFCS level I, 96 (48%) at GMFCS II, and 72 (36%) and 26 (13%) at GMFCS levels III and IV, respectively. All children received BoNT-A injections and participated in individual physiotherapy programs, as described in the Materials and Methods section. No treatment-related serious adverse events (SAEs) were observed during the observation period. The OGS total scores grew during the observation period (*p* < 0.001), with a significant difference in the results obtained between the first and the sixth assessment (*p* < 0.001) for both lower limbs (Table 1). The right and left lower limbs improved in 149 (74.5%) and 153 (76.5%) patients, respectively, and worsened in 17 (8.5%) and 14 (7%). The majority of children improved from one to four points (Table 2).

Initial foot contact (OGS section 2) improved (*p* < 0.001) during the five-year observation period (Table 3). The number of patients with toe contact at initial contact (IC) decreased from 72 (36%) to 15 (7.5%) and 70 (35%) to 18 (9%), and the number of patients with forefoot and flat foot IC increased from 110 (55%) to 144 (72%) and 109 (54.5%) to 138 (69%), and from 18 (9%) to 40 (20%) and 21 (10.5%) to 43 (21.5%), for the right and left feet, respectively.

This study showed increased scores for foot contact at MSt (OGS section 3) (*p* < 0.001) (Table 3). The number of patients who walked on their toes at MSt decreased from 128 (64%) to 79 (39.5%) for the right foot and from 128 (64%) to 75 (37.5%) for the left. After 5 years, more patients showed full foot contact, both right and left, throughout MSt, with an increase from 60 (30%) to 101 (50.5%) and from 59 (29.5%) to 103 (51.5%), respectively. Also, the timing of the heel rise (OGS section 4) improved (*p* < 0.001) (Table 3). The number of patients with no heel contact steadily decreased from 123 (61.5%) to 73 (36.5%) and from 122 (61%) to 70 (35%) for the right and left feet, respectively. The opposite trend was observed for the number of patients whose heels raised at early and late MSt. In the first case, the increase over the total observation period was 26 (13%) for the right lower limb and 28 (14%) for the left limb, while in the second case, it was 18 (9%) for both the right and left lower limbs.

The number of patients with limited or no foot clearance decreased from 176 (88%) to 165 (82.5%) and 178 (89%) to 167 (83.5%) for the right and left feet, respectively (Table 3).

The number of children with knee hyperextension at MSt decreased from 19 (9.5%) to 8 (4%) for the right knee joint and from 22 (11%) to 7 (3.5%) for the left (Table 3). An increase in patients who walked with knee flexion was observed, but flexion did not exceed 10 degrees for most. We observed 33 (16.5%) and 32 (16%) during the first assessment and 43 (21.5%) and 41 (20.5%) during the last assessment for the right and left knee joints, respectively. The number of patients walking with knee flexion greater than 15 degrees increased by 2.5%. Interestingly, the number of patients with right and left knee extension at TSt increased from 61 (30.5%) to 102 (51%) (*p* < 0.001) (Table 3).

The number of patients with knee flexion of less than 15 degrees during TSw decreased from 70 (35%) and 73 (36.5%) for the right and left knees to 56 (28%) for both limbs (Table 3). At the same time, the number of patients with right and left knee flexion greater than 30 degrees increased from 7 (3.5%) and 6 (3%) to 19 (9.5%) and 22 (11%) for the right and left lower limbs, respectively.

The hip joint position at TSt showed an improvement in extension from 4 (2%) and 3 (1.5%) to 18 (9%) and 16 (8%) patients, or to a neutral position from 111 (55.5%), 110 (55%) to 122 (61%), 122 (61%) for the right and left hip joints (*p* = 0.0012, *p* = 0.0014) (Table 3).

## 3. Discussion

The results presented are part of a more extensive study. A previous publication [7] showed improved gait-related activities in most children, with shifts in their GMFCS levels and FMS scores. Our main finding from this part is that treatment markedly improved their gait kinematics. The improvement was stable and even increased over 5 years.

The first goal of this study was to evaluate the influence of the BoNT-A combined with individualized goal-oriented therapy on gait patterns, measured by the OGS. The results show an increase in the total OGS score in the following years and a significant difference between the first and sixth measurements. Improvement after 5 years of follow-up occurred in 74.5% and 76.5% patients, and deterioration occurred in 8.5% and 7% of patients in the right and left lower extremities, respectively. The changes in the OGS typically ranged from 1 to 4 points. This is the most extensive pediatric CP study to include the OGS as an outcome measure and to follow over 5 years of repeat BoNT-A injections and intensive rehabilitation. Dursun et al. [10] studied the effectiveness of BoNT-A in the treatment of gait disorders in 241 patients with CP (2–17 years) over a 52-month follow-up period (with 12 weeks of a blinded period). The authors showed a significant improvement in the total OGS score in patients treated with BoNT-A. Juneja et al. [11] evaluated the effect of long-term treatment (42 months of follow-up) BTX-A in a small heterogeneous group of CP patients and found a statistically significant improvement in the total OGS score by at least 2 points in 61% of patients. Similar results were reported by Dursun et al. [12], showing significant increases in OGS scores at weeks 4 and 12 of multi-profile treatment using BoNT-A, serial casts, and physiotherapy.

The second goal of this study was to evaluate changes in the gait kinematics of the foot, knee, and hip. A significant improvement was observed in measures related to the foot’s position in the stance (OGS sections 2, 3, and 4). The number of patients in whom the first contact of the foot was with the toes decreased from 36% to 7.5% and from 35% to 9% for the right and left feet, respectively. The number of patients who experienced early heel rise at MSt decreased from 64% to 39.5% for the right foot and 64% to 37.5% for the left foot. The number of patients with no heel contact in the stance decreased steadily from 61.5% and 61% to 36.5% and 35%, respectively, for the right and left feet. The OGS is valid and reliable, with acceptable inter-rater and intra-rater reliability for the first four items [13]. In the abovementioned study, Dursun et al. [10] showed significant improvements by at least one grade at weeks 4 and 12 of BoNT-A treatment in 43–47 patients, respectively. No other papers in the literature show the use of OGS on a similar group and a similar observation period. Still, studies are available in which the authors have used the Physician Rating Scale (PRS), which assesses key events during the gait cycle with a structure similar to the OGS. Cardoso et al. [14] studied the effects of BoNT-A based on a review of 183 patients with excessive plantar flexion. The authors found a statistically significant improvement in PRS compared to a placebo. However, the period of observation and the mixed study groups do not allow for direct comparison. A study by Koman et al. [15] showed improvement in the PRS in 55% (*n* = 63/115) of patients treated with repeated injections of BoNT-A during the first year of follow-up. The authors reported improved gait pattern and hindfoot position in 44% (*n* = 20/45) maintained for 2 years. The spasticity of plantar flexors and weaknesses of pretibial muscles led to insufficient foot lift during a swing. Most patients had limited foot clearance (FC) in the swing phase. The study showed a slight decrease in these patients during the five-year follow-up period. However, most patients used AFOs that blocked foot drop and improved FC. Excessive flexion of the knee joint during the stance period increases energy expenditure and overload in the knee joint [16]. If left untreated, it leads to pain in the knee joints and lower spine, deformities, and osteoarticular diseases, contributing to limitations of gait function in adulthood [17,18,19]. Most of the patients in the study group presented a neutral knee position at MSt throughout the follow-up period. This is typical for patients 2–3 years of age, but with development, there is usually an increase in knee flexion [20]. The remaining patients walked with hyperextension or excessive knee flexion. The number of patients with a hyperextended knee decreased from 9.5% to 4% for the right knee joint and from 11% to 3.5% for the left knee joint. Patients walking with the right and left knees flexed increased from 16.5% and 16% to 21.5% and 20.5%, respectively, but the flexion value did not exceed 10 degrees. Only five patients developed severe flexion gait patterns (knee flexion > 15 degrees). Knee and hip extension at TSt and proper hip flexion and full knee extension during TSw in the opposite limb determine the appropriate stride length. The position of the knee joint at TSt was also examined. Over a five-year follow-up period, the number of patients waking with knee extension increased from 61 (30.5%) to 102 (51%). At the same time, the number of children with flexed hip joints at TSt decreased from 42.5% and 43.5% to 30% and 31% for the right and left sides, respectively. The last parameter assessed was the position of the knee joint during TSw. Throughout the follow-up period, the largest group of patients had knee flexion within the range of 16 to 30 degrees. The number of patients with knees flexed more than 30 degrees increased from 3.5% and 3% to 9.5% and 11% for the right and left limbs, respectively.

In summary, our study showed a significant reduction in kinematic abnormalities in a significant percentage of patients. The available literature presents the progression of gait disorders with age. Johnson et al. [21] described a deterioration in the parameters of kinematic effects related to the sagittal plain in 18 patients followed for 32 months. Bell et al. [22] drew similar conclusions based on over four years of observation of 28 patients. In a review based on four studies involving a total of 75 patients with CP, O’Sullivan et al. [18] demonstrated knee flexion deformity progression in all four papers. The authors also emphasize the progressive nature of knee flexion (extension limitation) at TSt. Wren et al. [23] analyzed the distribution of gait disorders in a group of nearly 500 patients with CP. Although the group was heterogeneous, the authors provided detailed characteristics of gait abnormalities in a subgroup of individuals with diplegia who had not undergone prior surgical treatment. Considering the objectives of this study, it is appropriate to compare the gait disorders described by the authors with the abnormalities observed in our study group during the last (sixth) assessment. Both groups show similar numbers: 200 and 158, with mean ages of 7.6 (±0.6) and 7.7 (±3.2), including patients who received only conservative treatment, representing the so-called natural history of CP. According to the authors, the proportion of patients who walked on their toes was 69%, significantly higher than the results from our study (39.5% and 37.5% for the right and left lower limbs, respectively). Additionally, the incidence of walking with flexed knees was reported at 78%, compared to about 25% in the present study. However, it is important to note the authors’ point that the patients experienced such severe gait disorders that they qualified for gait analysis to determine the need for surgery or further conservative treatment. However, a similar assumption can be made in our study group.

## 4. Conclusions

Botulinum toxin type A has emerged as a vital therapeutic agent for managing spasticity, supported by ongoing research trends and rising clinical adoption [24]. It has previously been reported that in children with CP, long-term treatment with BoNT-A is generally well tolerated and improves muscle hypertonia, spasticity, and gross motor function [4,7]. The results of this study provide further evidence for the efficacy of the treatment in improving gait kinematics and altering the natural history of deformation development. Our observational data suggest that integrated, repeated BoNT-A treatment as a part of intensive rehabilitation is associated with long-term improvements. The strengths of this study include a homogenous group of young children with bilateral spastic cerebral palsy treated in one center with a standardized intensive therapy program, a large number of patients assessed, and a long duration of follow-up.

The limitations of this study include its retrospective design and the lack of a control group, which could potentially introduce bias. Another limitation is the absence of a standard of home therapy. This may be a weakness of our study. However, according to our guidelines, all patients continued therapy at home and in daycare units. One significant limitation of the study was the absence of 3D gait analysis, which is a golden standard for kinematic assessment. Because the study aimed to show the influence of integrated management on the natural history of gait problems, we decided to analyze data from very young children who are not the best candidates for 3D gait analysis. In their original report, Boyd et al. suggested that the OGS may be particularly useful when children are too small or are insufficiently cooperative for instrumented gait analysis [9]. Other strengths include the experienced therapists, who had extensive training in applying the assessments used in the study, and the gait lab-based video recordings in a standardized environment. The results of this study align with previously published findings that demonstrate improvements in mobility and gross motor function [7]. We advocate for and promote integrated rehabilitation that includes BoNT-A injections to enhance gross motor skills and mobility while preventing gait kinematic deterioration in children with spastic bilateral cerebral palsy. Future research should incorporate validated, patient-centered outcome measures emphasizing life satisfaction and quality of life. It will be essential to evaluate whether mobility improvements are sustained over extended periods, such as through adolescence and into adulthood.

## 5. Material and Methods

### 5.1. Settings and Inclusion Criteria

Reported here are the pre-specified gait-related efficacy results from a retrospective, single-center, observational study conducted at the Mazovian Neuropsychiatry Center in Zagórze, Poland. A homogeneous group of patients with cerebral palsy was examined in the Movement Analysis Laboratory at Mazovian Neuropsychiatry Center in Poland.

The inclusion criteria were as follows: cerebral palsy with bilateral spastic paresis, a Gross Motor Function Classification System (GMFCS) level from I to IV, a first gait analysis conducted between 13 and 46 months of age, availability of gait analysis over a 5-year follow-up period, gait analysis performed at least once every 12 months, and a gait study conducted either before or at least 3 months after botulinum toxin type A injection.

The exclusion criteria included other forms of cerebral palsy, GMFCS V due to the inability to perform reliable gait assessments, and children with a history of prior selective dorsal rhizotomy, single-event multi-level surgeries, or other types of orthopedic surgeries. Two hundred consecutive patients who met all the inclusion criteria and did not meet any exclusion criteria were enrolled in the study.

### 5.2. BoNT-A Treatment and Rehabilitation Program

During the follow-up period, all children underwent BoNT-A injections and an individual physiotherapy program and used ankle foot orthosis (AFO). BoNT-A was most often administered 1 to 2 times a year; most patients received 5 to 10 BoNT-A (mean 7.6 ± 2.3) treatments during the observation period. The total doses per session varied from 20 u/kg to 30 u/kg for ABOBoNT-A/Dysport (Ipsen Biopharm Ltd., Wrexham, UK) or from 10 u/kg to 20 u/kg for OnaBoNT-A/ Botox (Allergan an AbbVie company, Irvine, CA, USA). In the majority, multilevel injections were performed. Hip flexors and adductors, knee flexors, and foot plantar flexors were usually selected based on detailed assessment, including muscle tone, range of motion, strength, and gait analysis. Injections were administered under mild sedation with midazolam, and electrical stimulation or ultrasound guidance was used. None of the patients received oral therapies for spasticity during the observation period. A physiotherapy program was planned individually for each child. Functional goals were outlined to organize the therapy process. Goals emphasizing functional mobility, including daily living activities, were developed with children and their parents. These goals adhered to the five SMART principles: specific, measurable, achievable, realistic, and time-bound [25,26]. The long-term goals were designed to enhance the child’s mobility and independence in daily activities and related social participation. The short-term goals centered on the body structure level and included improving passive and active range of motion, muscle strength, and selective motor control [27,28]. The frequency of intensive physiotherapy periods ranged from 3 to 12 weeks per year. Physiotherapy included individual and group training, lasting an average of 120 min daily. Physiotherapists used analytical therapy techniques, as well as functional and task-oriented training approaches. All assessed children continued rehabilitation on a community basis, conducted 2 to 5 times per week. All patients used a rigid or semirigid AFO, along with the proper footwear. The AFOs were tuned individually based on ground reaction vector visualization. The purpose of orthotic management was to improve the gait parameters and movement patterns. A total of 169 (84.5%) children wore an AFO from 5 to 8 h per day, while 31 (15.5%) wore one less than 5 h daily.

### 5.3. Data Collection Procedure

Information on BoNT-A treatment, casting, physiotherapy, orthoses, and GMFCS levels was collected from medical records. The OGS and kinematics were assessed using two-plane clinical video recordings conducted in the same gait laboratory for all children.

The video recording process included bright lighting and an appropriate contrast background. Cameras were positioned perpendicularly to the assessed planes of motion, maintaining enough distance to minimize optical errors. The walkway path measured 10 m long and 2 m wide, while the visible measurement area on the screen was 5 m by 1.6 m. Patient preparation for the assessment required exposing the lower body and walking barefoot. According to the inclusion criteria, all 200 children completed a minimum of six gait assessments over a five-year follow-up period, with suitable intervals of approximately one year. The data from assessments conducted either before or at least three months after the injection of BoNT-A were analyzed over the five years. Six gait assessments were selected for each patient. All assessors involved in the study were well-trained and had extensive gait analysis experience. The measurement results were presented using the Observation Gait Scale (OGS). The OGS consists of eight sections: (1) knee position at midstance (MSt), (2) initial foot contact, (3) foot contact at midstance, (4) timing of heel rise, (5) hindfoot at midstance, (6) base of support, (7) gait assistive devices, and (8) change. Scoring was carried out for both the left and right lower extremities by selecting the appropriate numerical value. A perfect score is 22 for each limb. Lower scores indicate greater gait impairments; the higher the score, the fewer impairments the child exhibits [9]. The first four sections of the OGS were utilized to analyze kinematic changes in gait. To provide a more precise description of sagittal plane gait kinematics, these items were complemented with an assessment of the knee and hip joint positions at terminal stance (TSt), knee joint position during terminal swing (TSw), and foot clearance during swing (Sw). Numerical values similar to those used in the OGS were assigned to these measurements to facilitate data collection and evaluation (Table 3). Due to this study’s retrospective nature, which involved analyzing medical records, no application was submitted to the Bioethics Committee for research and scientific use of the results obtained. The authors (MB, WP) are employees of the Neurorehabilitation Department and have previously treated the assessed group.

### 5.4. Statistical Analysis

The outcomes were summarized descriptively, including the N, median, 25th and 75th percentiles, minimum, and maximum. All calculations were performed using the R statistical package, version 3.6.0. The eighth section, which determines change, was excluded from the statistical analysis. The maximum total score achieved and the results from the first four sections of the scale provide the foundation for further statistical analysis. Due to the requirements of the statistical package used for the analysis, the detailed scores for the MSt foot and knee position in the first four OGS sections were modified to eliminate any duplicate values (e.g., 0 and 0 for different positions of the same joint) (Table 3). Changed values were used only for kinematic description. This adjustment did not impact the total OGS score calculation, as the original scoring was maintained for this part. The significance level for the total OGS scores was calculated using the Kruskal–Wallis and Mann–Whitney U tests (*p* = 0.05). Due to repeated observations of the same patient, mixed models were employed to assess the relationship between the results of gait kinematics during follow-up.

## Figures and Tables

**Table 1 toxins-17-00142-t001:** Total OGS scores during 5-year follow-up period.

Assessment Number													
Total Max 22	1 R (N = 200)	1 L (N = 200)	2 R (N = 200)	2 L (N = 200)	3R (N = 200)	3 L (N = 200)	4R (N = 200)	4 L (N = 200)	5R (N = 200)	5 L (N = 200)	6R (N = 200)	6 L (N = 200)	Test
**2**	0% (N = 0)	0% (N = 0)	0% (N = 0)	0% (N = 0)	0% (N = 0)	0% (N = 0)	0% (N = 0)	0% (N = 0)	0,5% (N = 1)	0,5% (N = 1)	0,5% (N = 1)	0,5% (N = 1)	
**3**	3,5% (N = 7)	3,5% (N = 7)	1% (N = 2)	1% (N = 2)	1% (N = 2)	1% (N = 2)	1% (N = 2)	1% (N = 2)	0,5% (N = 1)	0,5% (N = 1)	0,5% (N = 1)	0,5% (N = 1)	
**4**	8,5% (N = 17)	8,5% (N = 17)	2,5% (N = 5)	2,5% (N = 5)	1,5% (N = 3)	1,5% (N = 3)	1,5% (N = 3)	1,5% (N = 3)	1,5% (N = 3)	1,5% (N = 3)	1,5% (N = 3)	2% (N = 4)	
**5**	6,5% (N = 13)	6,5% (N = 13)	7% (N = 14)	7% (N = 14)	3,5% (N = 7)	4% (N = 8)	2,5% (N = 5)	3% (N = 6)	2% (N = 4)	2% (N = 4)	2,5% (N = 5)	2% (N = 4)	
**6**	5,5% (N = 11)	5,5% (N = 11)	5,5% (N = 11)	6% (N = 12)	3,5% (N = 7)	3% (N = 6)	3% (N = 6)	2,5% (N = 5)	5% (N = 10)	4,5% (N = 9)	4% (N = 8)	4% (N = 8)	
**7**	8,5% (N = 17)	8,5% (N = 17)	5% (N = 10)	4,5% (N = 9)	7% (N = 14)	7% (N = 14)	4% (N = 8)	4% (N = 8)	2,5% (N = 5)	3,5% (N = 7)	2,5% (N = 5)	2,5% (N = 5)	
**8**	6,5% (N = 13)	6% (N = 12)	6% (N = 12)	5,5% (N = 11)	4% (N = 8)	4% (N = 8)	6,5% (N = 13)	5,5% (N = 11)	5,5% (N = 11)	4% (N = 8)	4% (N = 8)	4% (N = 8)	
**9**	11,5% (N = 23)	12% (N = 24)	11% (N = 22)	11% (N = 22)	9,5% (N = 19)	9,5% (N = 19)	7% (N = 14)	7,5% (N = 15)	5,5% (N = 11)	7% (N = 14)	7% (N = 14)	7% (N = 14)	
**10**	10,5% (N = 21)	9% (N = 18)	12% (N = 24)	11,5% (N = 23)	11% (N = 22)	10,5% (N = 21)	11,5% (N = 23)	11% (N = 22)	10,5% (N = 21)	10% (N = 20)	11,5% (N = 23)	9,5% (N = 19)	
**11**	12% (N = 24)	12,5% (N = 25)	9,5% (N = 19)	9,5% (N = 19)	12,5% (N = 25)	12% (N = 24)	13% (N = 26)	12% (N = 24)	12,5% (N = 25)	12% (N = 24)	12% (N = 24)	12,5% (N = 25)	
**12**	10% (N = 20)	10% (N = 20)	12,5% (N = 25)	11% (N = 22)	12% (N = 24)	10,5% (N = 21)	13% (N = 26)	13% (N = 26)	13,5% (N = 27)	12% (N = 24)	14,5% (N = 29)	13% (N = 26)	
**13**	9% (N = 18)	9% (N = 18)	12% (N = 24)	14% (N = 28)	15,5% (N = 31)	15,5% (N = 31)	16% (N = 32)	15% (N = 30)	15% (N = 30)	16,5% (N = 33)	14% (N = 28)	16% (N = 32)	
**14**	5% (N = 10)	6,5% (N = 13)	9,5% (N = 19)	10,5% (N = 21)	10,5% (N = 21)	12,5% (N = 25)	10,5% (N = 21)	11,5% (N = 23)	13% (N = 26)	12,5% (N = 25)	12,5% (N = 25)	12,5% (N = 25)	
**15**	2% (N = 4)	1% (N = 2)	4% (N = 8)	4% (N = 8)	5,5% (N = 11)	5% (N = 10)	7% (N = 14)	8% (N = 16)	7,5% (N = 15)	8% (N = 16)	8% (N = 16)	8,5% (N = 17)	
**16**	1% (N = 2)	1% (N = 2)	2,5% (N = 5)	1,5% (N = 3)	3% (N = 6)	3% (N = 6)	3% (N = 6)	3% (N = 6)	4% (N = 8)	4% (N = 8)	4% (N = 8)	4% (N = 8)	
**17**	0% (N = 0)	0,5% (N = 1)	0% (N = 0)	0,5% (N = 1)	0% (N = 0)	1% (N = 2)	0,5% (N = 1)	1,5% (N = 3)	1% (N = 2)	1,5% (N = 3)	1% (N = 2)	1,5% (N = 3)	
**Median**	9 (7–12)	9 (7–12)	10,5 (8–13)	11 (9–13)	11 (9–13)	11 (9–13)	11,5 (9–13)	12 (10–14)	12 (10–14)	12 (10–14)	12 (10–14)	12 (10–14)	Kruskal–Wallis (***p* < 0,001**)
**Range**	3–16	3–17	3–16	3–17	3–16	3–17	3–17	3–17	2–17	2–17	2–17	2–17	
**Comparison of the total OGS scores achieved during the first and sixth assessments.**													Mann–Whitney U (***p* < 0,001**)

R—right lower limb; L—left lower limb. Scores are presented as medians (25th and 75th percentiles).

**Table 2 toxins-17-00142-t002:** Changes in total OGS scores during the 5-year follow-up period.

Change in Total OGS Score	Right Lower Limb N = 200	Left Lower Limb N = 200
**−3**	0,5% (N = 1)	0,5% (N = 1)
**−2**	3% (N = 6)	2% (N = 4)
**−1**	5% (N = 10)	4,5% (N = 9)
**0**	15% (N = 30)	15% (N = 30)
**1**	17% (N = 34)	17,5% (N = 35)
**2**	18% (N = 36)	18% (N = 36)
**3**	15,5% (N = 31)	17% (N = 34)
**4**	10,5% (N = 21)	11,5% (N = 23)
**5**	8% (N = 16)	6,5% (N = 13)
**6**	4% (N = 8)	4% (N = 8)
**7**	1,5% (N = 3)	2% (N = 4)

**Table 3 toxins-17-00142-t003:** Comparison of gait kinematic parameters during 5-year follow-up period.

Assessment Number															
	Gait Pattern (Score)	1R (N = 200)	1L (N = 200)	2R (N = 200)	2L (N = 200)	3R (N = 200)	3L (N = 200)	4R (N = 200)	4L (N = 200)	5R (N = 200)	5L (N = 200)	6R (N = 200)	6L (N = 200)	Test R	Test L
**Initial Foot Contact**	Toe (0)	36% (N = 72)	35% (N = 70)	18% (N = 36)	17,5% (N = 35)	14% (N = 28)	13,5% (N = 27)	11,5% (N = 23)	11,5% (N = 23)	9,5% (N = 19)	9,5% (N = 19)	7,5% (N = 15)	9% (N = 18)		
	Forefoot (1)	55% (N = 110)	54,5% (N = 109)	70,5% (N = 141)	68,5% (N = 137)	73% (N = 146)	71% (N = 142)	71,5% (N = 143)	68,5% (N = 137)	70,5% (N = 141)	69% (N = 138)	72% (N = 144)	69% (N = 138)		
	Flat Foot (2)	9% (N = 18)	10,5% (N = 21)	11,5% (N = 23)	14% (N = 28)	13% (N = 26)	15% (N = 30)	16,5% (N = 33)	19% (N = 38)	19,5% (N = 39)	21% (N = 42)	20% (N = 40)	21,5% (N = 43)		
	Heel (3)	0% (N = 0)	0% (N = 0)	0% (N = 0)	0% (N = 0)	0% (N = 0)	0,5% (N = 1)	0,5% (N = 1)	1% (N = 2)	0,5% (N = 1)	0,5% (N = 1)	0,5% (N = 1)	0,5% (N = 1)		
	Median	1 (0–1)	1 (0–1)	1 (1–1)	1 (1–1)	1 (1–1)	1 (1–1)	1 (1–1)	1 (1–1)	1 (1–1)	1 (1–1)	1 (1–1)	1 (1–1)	Kruskal–Wallis (***p* < 0,001**)	Kruskal–Wallis (***p* < 0,001**)
	Range	0–2	0–2	0–2	0–2	0–2	0–3	0–3	0–3	0–3	0–3	0–3	0–3		
**Comparison of the scores for initial foot contact achieved during the first and sixth assessments.**														Mann–Whitney U (***p* < 0,001**)	Mann–Whitney U (***p* < 0,001**)
**Foot Contact at Midstance**	Toe/Toe (0)	64% (N = 128)	64% (N = 128)	52,5% (N = 105)	51% (N = 102)	47,5% (N = 95)	46% (N = 92)	42,5% (N = 85)	40,5% (N = 81)	39,5% (N = 79)	38,5% (N = 77)	39,5% (N = 79)	37,5% (N = 75)		
	Flat Foot/Early Heel Rise (1)	30% (N = 60)	29,5% (N = 59)	41% (N = 82)	41,5% (N = 83)	46,5% (N = 93)	46,5% (N = 93)	50% (N = 100)	50,5% (N = 101)	51% (N = 102)	51,5% (N = 103)	50,5% (N = 101)	51,5% (N = 103)		
	Flat Foot/No Early Heel Rise (2)	6% (N = 12)	6,5% (N = 13)	6,5% (N = 13)	7,5% (N = 15)	6% (N = 12)	7,5% (N = 15)	7,5% (N = 15)	9% (N = 18)	9,5% (N = 19)	10% (N = 20)	10% (N = 20)	11% (N = 22)		
	Median	0 (0–1)	0 (0–1)	0 (0–1)	0 (0–1)	1 (0–1)	1 (0–1)	1 (0–1)	1 (0–1)	1 (0–1)	1 (0–1)	1 (0–1)	1 (0–1)	Kruskal–Wallis (***p* < 0,001**)	Kruskal–Wallis (***p* < 0,001**)
	Range	0–2	0–2	0–2	0–2	0–2	0–2	0–2	0–2	0–2	0–2	0–2	0–2		
**Comparison of the scores for foot contact at midstance achieved during the first and sixth assessments.**														Mann–Whitney U (***p* < 0,001**)	Mann–Whitney U (***p* < 0,001**)
**Timing of Heel Rise**	No Heel Rise (−1)	3% (N = 6)	3% (N = 6)	3% (N = 6)	3,5% (N = 7)	2,5% (N = 5)	3% (N = 6)	4,5% (N = 9)	5% (N = 10)	5% (N = 10)	5% (N = 10)	5,5% (N = 11)	5,5% (N = 11)		
	No Heel Contact (0)	61,5% (N = 123)	61% (N = 122)	49% (N = 98)	48,2% (N = 96)	45% (N = 90)	44% (N = 88)	40,2% (N = 80)	38,5% (N = 77)	37% (N = 74)	36% (N = 72)	36,5% (N = 73)	35% (N = 70)		
	Before 25% Stance (1)	22% (N = 44)	21% (N = 42)	30% (N = 60)	28,6% (N = 57)	31,5% (N = 63)	30% (N = 60)	35,2% (N = 70)	34% (N = 68)	35% (N = 70)	35% (N = 70)	35% (N = 70)	35% (N = 70)		
	Between 25 and 50% Stance (2)	12,5% (N = 25)	13,5% (N = 27)	17% (N = 34)	18,1% (N = 36)	20% (N = 40)	21,5% (N = 43)	19,1% (N = 38)	21% (N = 42)	21,5% (N = 43)	22% (N = 44)	21,5% (N = 43)	22,5% (N = 45)		
	At Terminal Stance (3)	1% (N = 2)	1,5% (N = 3)	1% (N = 2)	1,5% (N = 3)	1% (N = 2)	1,5% (N = 3)	1% (N = 2)	1,5% (N = 3)	1,5% (N = 3)	2% (N = 4)	1,5% (N = 3)	2% (N = 4)		
	Median	0 (0–1)	0 (0–1)	0 (0–1)	0 (0–1)	1 (0–1)	1 (0–1)	1 (0–1)	1 (0–1)	1 (0–1)	1 (0–1)	1 (0–1)	1 (0–1)	Chi-squared (*p* = 0,0065)	Chi-squared (***p* = 0,0037)**
	Range	−1–3	−1–3	−1–3	−1–3	−1–3	−1–3	−1–3	−1–3	−1–3	−1–3	−1–3	−1–3		
**Comparison of the scores for the timing of heel rise achieved during the first and sixth assessments.**														Mann–Whitney U (***p* = 0,0001**)	Mann–Whitney U (***p* = 0,0001**)
**Foot Clearance**	Poor (0)	88% (N = 176)	89% (N = 178)	88,5% (N = 177)	88,5% (N = 177)	87,5% (N = 175)	87% (N = 174)	86% (N = 172)	85% (N = 170)	84,5% (N = 169)	83,5% (N = 167)	82,5% (N = 165)	83,5% (N = 167)	Chi-squared (*p* = 0,4796)	Chi-squared (*p* = 0,4208)
	Normal (1)	12% (N = 24)	11% (N = 22)	11,5% (N = 23)	11,5% (N = 23)	12,5% (N = 25)	13% (N = 26)	14% (N = 28)	15% (N = 30)	15,5% (N = 31)	16,5% (N = 33)	17,5% (N = 35)	16,5% (N = 33)		
**Comparison of the scores for foot clearance achieved during the first and sixth assessments.**														Fisher (***p* < 0,001**)	Fisher (***p* < 0,001**)
**Knee Position at Midstance**	Knee Hyperextension 5–10° (−2)	0,5% (N = 1)	0,5% (N = 1)	0% (N = 0)	0% (N = 0)	0% (N = 0)	0% (N = 0)	0% (N = 0)	0% (N = 0)	0% (N = 0)	0% (N = 0)	0% (N = 0)	0% (N = 0)		
	Knee Hyperextension < 5° (−1)	9% (N = 18)	10,5% (N = 21)	9,5% (N = 19)	11% (N = 22)	9% (N = 18)	8,5% (N = 17)	7% (N = 14)	6,5% (N = 13)	5% (N = 10)	4,5% (N = 9)	4% (N = 8)	3,5% (N = 7)		
	Normal (0)	63% (N = 126)	62% (N = 124)	63% (N = 126)	61,5% (N = 123)	65,5% (N = 131)	65% (N = 130)	63% (N = 126)	64% (N = 128)	61,5% (N = 123)	62,5% (N = 125)	63% (N = 126)	63% (N = 126)		
	Knee Flexion < 10° (1)	16,5% (N = 33)	16% (N = 32)	20,5% (N = 41)	19,5% (N = 39)	17,5% (N = 35)	17,5% (N = 35)	21% (N = 42)	20% (N = 40)	22,5% (N = 45)	21% (N = 42)	21% (N = 42)	20,5% (N = 41)		
	Knee Flexion 10–15° (2)	9,5% (N = 19)	9,5% (N = 19)	6% (N = 12)	7% (N = 14)	6,5% (N = 13)	7,5% (N = 15)	7% (N = 14)	7,5% (N = 15)	7,5% (N = 15)	8,5% (N = 17)	8% (N = 16)	9% (N = 18)		
	Knee Flexion > 15° (3)	1,5% (N = 3)	1,5% (N = 3)	1% (N = 2)	1% (N = 2)	1,5% (N = 3)	1,5% (N = 3)	2% (N = 4)	2% (N = 4)	3,5% (N = 7)	3,5% (N = 7)	4% (N = 8)	4% (N = 8)		
	Median	0 (0–1)	0 (0–1)	0 (0–1)	0 (0–1)	0 (0–1)	0 (0–1)	0 (0–1)	0 (0–1)	0 (0–1)	0 (0–1)	0 (0–1)	0 (0–1)	Kruskal– Wallis (*p* = 0,0793)	Kruskal–Wallis (***p* = 0,0502)**
	Range	−2–3	−2–3	−1–3	−1–3	−1–3	−1–3	−1–3	−1–3	−1–3	−1–3	−1–3	−1–3		
**Comparison of the scores for knee position at midstance achieved during the first and sixth assessments.**														Mann–Whitney U (*p* = 0,0621)	Mann–Whitney U (***p* = 0,0233**)
**Knee Position at Terminal Stance**	Knee Flexion (0)	69,5% (N = 139)	69,5% (N = 139)	61,5% (N = 123)	63% (N = 126)	52,5% (N = 105)	54% (N = 108)	50% (N = 100)	52% (N = 104)	48,5% (N = 97)	48,5% (N = 97)	49% (N = 98)	49% (N = 98)		
	Knee Extension (1)	30,5% (N = 61)	30,5% (N = 61)	38,5% (N = 77)	37% (N = 74)	47,5% (N = 95)	46% (N = 92)	50% (N = 100)	48% (N = 96)	51,5% (N = 103)	51,5% (N = 103)	51% (N = 102)	51% (N = 102)		
	Median	0 (0–1)	0 (0–1)	0 (0–1)	0 (0–1)	0 (0–1)	0 (0–1)	0,5 (0–1)	0 (0–1)	1 (0–1)	1 (0–1)	1 (0–1)	1 (0–1)	Kruskal– Wallis (***p* < 0,001)**	Kruskal–Wallis (***p* < 0,001)**
	Range	0–1	0–1	0–1	0–1	0–1	0–1	0–1	0–1	0–1	0–1	0–2	0–1		
**Comparison of the scores for knee position at terminal stance achieved during the first and sixth assessments.**														Mann–Whitney U (***p* < 0,001**)	Mann–Whitney U (***p* < 0,001**)
**Knee Position during Terminal Swing**	Knee Flexion > 45° (0)	1,5% (N = 3)	1% (N = 2)	1% (N = 2)	0,5% (N = 1)	0,5% (N = 1)	0,5% (N = 1)	1% (N = 2)	0,5% (N = 1)	1% (N = 2)	0,5% (N = 1)	1% (N = 2)	1% (N = 2)		
	Knee Flexion 31–45° (1)	2% (N = 4)	2% (N = 4)	1,5% (N = 3)	1,5% (N = 3)	3% (N = 6)	3% (N = 6)	4% (N = 8)	3,5% (N = 7)	7,5% (N = 15)	8% (N = 16)	8,5% (N = 17)	10% (N = 20)		
	Knee Flexion 16–30° (2)	61,5% (N = 123)	60,5% (N = 121)	63% (N = 126)	62,5% (N = 125)	67,5% (N = 135)	67% (N = 134)	66,5% (N = 133)	68% (N = 136)	64% (N = 128)	65,5% (N = 131)	62,5% (N = 125)	61% (N = 122)		
	Knee Extension or Knee Flexion ≤ 15° (3)	35% (N = 70)	36,5% (N = 73)	34,5% (N = 69)	35,5% (N = 71)	29% (N = 58)	29,5% (N = 59)	28,5% (N = 57)	28% (N = 56)	27,5% (N = 55)	26% (N = 52)	28% (N = 56)	28% (N = 56)		
	Median	2 (2–3)	2 (2–3)	2 (2–3)	2 (2–3)	2 (2–3)	2 (2–3)	2 (2–3)	2 (2–3)	2 (2–3)	2 (2–3)	2 (2–3)	2 (2–3)	Kruskal– Wallis (*p* = 0,0755)	Kruskal–Wallis (***p* = 0,0078)**
	Range	0–3	0–3	0–3	0–3	0–3	0–3	0–3	0–3	0–3	0–3	0–3	0–3		
**Comparison of the scores for knee position during terminal swing achieved during the first and sixth assessments.**														Mann–Whitney U (***p* = 0,0328**)	Mann–Whitney U (***p* = 0,0078**)
Hip Position at Terminal Stance	Hip Flexion (0)	42,5% (N = 85)	43,5% (N = 87)	35,5% (N = 71)	37% (N = 74)	28% (N = 56)	29,5% (N = 59)	31% (N = 62)	32% (N = 64)	30,5% (N = 61)	31,5% (N = 63)	30% (N = 60)	31% (N = 62)		
	Hip in Neutral (1)	55,5% (N = 111)	55% (N = 110)	62% (N = 124)	61% (N = 122)	70% (N = 140)	68,5% (N = 137)	65% (N = 130)	64,5% (N = 129)	61% (N = 122)	60,5% (N = 121)	61% (N = 122)	61% (N = 122)		
	Hip Extension (2)	2% (N = 4)	1,5% (N = 3)	2,5% (N = 5)	2% (N = 4)	2% (N = 4)	2% (N = 4)	4% (N = 8)	3,5% (N = 7)	8,5% (N = 17)	8% (N = 16)	9% (N = 18)	8% (N = 16)		
	Median	1 (0–1)	1 (0–1)	1 (0–1)	1 (0–1)	1 (0–1)	1 (0–1)	1 (0–1)	1 (0–1)	1 (0–1)	1 (0–1)	1 (0–1)	1 (0–1)	Kruskal– Wallis (***p* = 0,0051)**	Kruskal–Wallis (***p* = 0,0054)**
	Range	0–2	0–2	0–2	0–2	0–2	0–2	0–2	0–2	0–2	0–2	0–2	0–2		
**Comparison of the scores for hip position at terminal stance achieved during the first and sixth assessments.**														Mann–Whitney U (***p* = 0,0012**)	Mann–Whitney U (***p* = 0,0014**)

R—right lower limb, L—left lower limb. Scores are presented as medians (25th and 75th percentiles).

## Data Availability

The original contributions presented in this study are included in this article. Further inquiries can be directed to the corresponding author.

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
