# Peer review of "Long-Term Improvement of Gait Kinematics in Young Children with Cerebral Palsy Treated with Botulinum Toxin Injections and Integrated/Intensive Rehabilitation: A 5-Year Retrospective Observational Study"

_toxins, 2025, doi:10.3390/toxins17030142_

Round 1
Reviewer 1 Report
Comments and Suggestions for Authors
In this study, the authors showed, through a retrospective study, the improvement in gait kinematic measurements and in preselected parameters related to the foot, knee, and hip in patients with cerebral palsy treated with repeated Botulinum Toxin A (BoNT-A) injections within a rehabilitation treatment over a five-year follow-up period.
Specific comments:
- The novelty of the study in comparison to the existing literature is unclear, as many studies cited in the discussion already demonstrate improved gait kinematics following BoNT-A injections combined with rehabilitation in children with cerebral palsy, in agreement with your study. Moreover, most of these studies often include long follow-up periods, large patient samples, and the use of scales such as the OGS or similar, much like your study.
- I believe that an important weak aspect of the study is the lack of a placebo group or at least a comparison between patients receiving integrated treatment (BoNT-A and rehabilitation) and those receiving either rehabilitation alone or BoNT-A alone. Without such comparisons, the reliability of the study may be questioned.
- I think it is necessary to specify whether the patient sample was homogeneous, as this enables proper stratification of patients. For example, it is not stated whether any children were receiving oral systemic therapy for spasticity or other surgical treatments. Additionally, does the inclusion of patients with different baseline levels of motor function (ranging from GMFCS level 1 to 4) not influence or impact the results? I believe this should be clarified.
- Why was Botulinum Toxin treatment administered only once or twice per year? Is this the standard regimen for pediatric patients? It would be helpful to clarify this by citing relevant literature. Furthermore, I think the evaluation conducted 3 months after the injection may be too late to assess the maximum effectiveness of the treatment.
- Have any side effects ever been reported since the initiation of botulinum toxin injections? Why is this not explicitly mentioned? How would they affect the results?
- I believe the Materials and Methods section in the abstract should be expanded, particularly by specifying for example the details of the Botulinum Toxin treatment protocol.
- The abbreviation GMFCS should be spelled out in full at least once in the text.
Author Response
Thank you very much for taking the time to review this manuscript. Please find the detailed responses below and the corresponding corrections highlighted in the re-submitted files.
Specific comments:
- Comments 1: The novelty of the study in comparison to the existing literature is unclear, as many studies cited in the discussion already demonstrate improved gait kinematics following BoNT-A injections combined with rehabilitation in children with cerebral palsy, in agreement with your study. Moreover, most of these studies often include long follow-up periods, large patient samples, and the use of scales such as the OGS or similar, much like your study.
- Response: Thank you for your feedback. I concur that the studies mentioned share similarities; however, neither examines a substantial homogeneous group of diplegic children nor spans a five-year observation period. Our study strongly addresses these gaps by emphasizing the necessity for evaluations of rehabilitation in real-world settings.
- Comments 2: I believe that an important weak aspect of the study is the lack of a placebo group or at least a comparison between patients receiving integrated treatment (BoNT-A and rehabilitation) and those receiving either rehabilitation alone or BoNT-A alone. Without such comparisons, the reliability of the study may be questioned.
- Response: Thank you for your comments. However, given the study's design, which includes 60 months of observations, maintaining a placebo group is not feasible from both practical and ethical standpoints.
- Comments 3: I think it is necessary to specify whether the patient sample was homogeneous, as this enables proper stratification of patients. For example, it is not stated whether any children were receiving oral systemic therapy for spasticity or other surgical treatments. Additionally, does the inclusion of patients with different baseline levels of motor function (ranging from GMFCS level 1 to 4) not influence or impact the results? I believe this should be clarified.
Response: Thank you for the correction. I added information regarding surgeries in the section on materials and methods, settings, and inclusion criteria. None of the patients used oral medication for spasticity, as it is not a popular method for that patient group in Poland. All patients were able to walk short distances with the help of walking devices and/or an accompanying person at a minimum.
- Comments 4: Why was Botulinum Toxin treatment administered only once or twice per year? Is this the standard regimen for pediatric patients? It would be helpful to clarify this by citing relevant literature. Furthermore, I think the evaluation conducted 3 months after the injection may be too late to assess the maximum effectiveness of the treatment.
- Response: Thank you for your question. For more than twenty years, we have aimed to administer BoNT A as infrequently as possible, averaging six months for the entire population. This strategy appears logical considering recent reports of potential muscle atrophy. The study's design sought to capture changes in gait that were not influenced by the direct denervation effect of the toxin, but rather by alterations in muscle and other tissue properties (such as muscle elongation, strength increases, etc.).
- Comments 5: Have any side effects ever been reported since the initiation of botulinum toxin injections? Why is this not explicitly mentioned? How would they affect the results?
- Response: Assessing treatment safety was not a goal of this analysis; we conducted over 10 RCTs in our center, focusing on the safety and efficacy of BoNT-A, many of which were published. As an injector in this study group, I can state that we never encountered treatment-related SAEs.
- Comments 6: I believe the Materials and Methods section in the abstract should be expanded, particularly by specifying for example the details of the Botulinum Toxin treatment protocol.
Response: Thank you for your feedback. I have elaborated on this section of the
abstract.
- Comments 7: The abbreviation GMFCS should be spelled out in full at least once in the text.
- Response: Thank you for the correction. I expanded the abbreviation.

Reviewer 2 Report
Comments and Suggestions for Authors
Title (Lines 1–4)
Consider clarifying that this is a retrospective observational study (per lines 11–12). For transparency, appending “... A 5-year Retrospective Observational Study” might increase clarity.
Abstract (Lines 6–27)
The abstract follows a structured approach (Introduction, Methods, Results, Conclusion), which is beneficial for clarity.
The conclusion highlights a significant, long-term positive effect. While that is valid, it would be good to explicitly recognize in the conclusion that this is observational data. Also emphasize that results may not be generalized to all clinical settings without caution.
Key Contribution (Lines 21–27)
The statement on lines 25–27 about “development of serious gait problems not observed” is quite strong. Clarify or temper this if the results only show that major deterioration was seen in only a small fraction, rather than “not observed” at all.
2. Introduction (Lines 29–52)
Lines 32–34: The text states that between 75 and 91% of CP patients have spasticity interfering with motor function. The cited study discusses the prevalence of spasticity in CP (Are 75-91 correct?), it doesn't specifically address whether this spasticity "interferes with motor function."
Role of Botulinum Toxin
Lines 35–40: The discussion of BoNT-A as an “important treatment modality” is well justified, highlighting its role in reducing hypertonia and preventing secondary complications. To further strengthen this section, the authors could also emphasize the growing research trends and increasing clinical adoption of BoNT-A for spasticity management (doi: 10.3390/toxins16040184).
The phrase “helping achieve maximal potential development” is somewhat broad;
Methods (Lines 53–124)
Lines 53–55: States that it is a single-center retrospective observational study at the Mazovian Neuropsychiatry Center in Poland.
Clarify exactly how retrospective data were extracted and if any standardized protocol existed during the data collection.
Inclusion Criteria
Since the study focuses on gait, it is logical to exclude GMFCS V. This is appropriate, though it might be helpful to state why GMFCS V was excluded (non-ambulatory, cannot perform reliable gait assessments).
Exclusion includes “other forms of CP, especially dyskinetic type,” GMFCS V, and orthopedic surgeries. You may also specify if children with prior rhizotomy or single-event multi-level surgeries (SEMLS) were excluded.
Lines 66–74: A yearly frequency (1–2 times), with total doses ABOBoNT-A 20–30 U/kg or OnabotulinumtoxinA 10–20 U/kg. Typically multi-level injections.
Specify whether the brand was chosen randomly or systematically, and if the dosing conversions between abobotulinumtoxinA and onabotulinumtoxinA were consistent with known ratio guidelines.
State if sedation protocols or local anesthetics were standardized.
Mention the average interval between injections (3–6 months?). Currently, it says 1 to 2 times a year, but the typical recommended minimum interval is ~3 months.
The rehabilitation intensity is described (3 to 12 weeks per year of “intensive” therapy, plus community-based therapy 2 to 5 times per week). However, the large variability in frequency (3–12 weeks) might introduce significant heterogeneity in outcomes. Consider discussing how you controlled or accounted for differences in therapy dose.
3.3 Data Collection Procedure
Lines 91–103: Clarify how standardization of video capture was ensured (e.g., were children walking barefoot or with AFOs? Same walkway length? Same camera angles?). This is crucial for consistent OGS scoring.
Indicate whether the OGS scorers were blinded to injection history or the timing of injections.
Lines 95–96: Six gait assessments are selected for each patient over 5 years, including pre- or at-least-3-months post-BoNT-A.This is logical, but are these time points standardized (e.g., 0, 12, 24, 36, 48, 60 months) or did it vary by patient? In retrospective studies, timing might differ widely if the child missed an appointment or received injections off schedule.
3.4 Statistical Analysis
Line 115: The text says “detailed scores for the MSt foot and knee position... have been modified to eliminate duplicate values.”
This might raise concerns. If you collapsed categories or changed scoring, it can alter validity. Clarify precisely which values were merged or recoded and why that does not affect the total OGS scoring.
Indicate how missing data (if any) were handled. Were any participants dropped from analysis if they missed one of the 6 visits?
4. Results (Lines 125–164)
Lines 125–128: You report 120 males, 80 females, with ages 24–46 months (mean 32.23 ± 6.96). GMFCS distribution is clearly stated: I (3%), II (48%), III (36%), IV (13%). Consider adding the exact n for each GMFCS level (e.g., n=6 for level I, n=96 for level II, etc.).
OGS Total Score
Lines 129–132: States total OGS scores “grew” over the period (p<0.001), with significant difference between the 1st and 6th assessments. Also indicates 74.5% and 76.5% improvement in right and left limbs, respectively, while 8.5% and 7% deteriorated. Adding median (or mean) OGS changes would further help quantify how large the improvement was. The tables do show median changes, but summarizing that in the text might be clearer.
Lines 133–164: You systematically describe the results for foot contact at IC, foot contact at midstance, timing of heel rise, knee position at midstance and terminal stance, hip position at terminal stance, etc. This thorough reporting is good; however, the text is quite dense. For clarity, you might reorganize so that each joint is summarized in a single paragraph (foot → knee → hip).
Mention some absolute numbers or percentages for each domain, plus the direction of change, which is partly done. For instance, lines 139–142 mention that toe-walking at midstance decreased from 64% to ~39%.
As it is a retrospective study, mention if any data points were missing. Did all 200 children complete all 6 assessments? The text implies yes, but please confirm.
5. Discussion & Conclusions (Lines 165–281)
Lines 250–260: The paper indicates that the improvements were stable or increased across the 5-year span. This addresses a common question about whether repeated BoNT-A leads to diminishing returns. Cite guidelines or systematic reviews that address repeated BoNT-A injections in CP.
Strengths and Limitations
Lines 261–269: Strengths include large sample size and standardized therapy center. Limitations include retrospective design, no control group, no 3D gait analysis.
The discussion of limitations is appropriate, but consider discussing how lack of randomization or standardized home therapy may have introduced confounding.Also mention the wide variance in physiotherapy “intensity” (3–12 weeks).
Lines 278–281: Concludes that integrated treatment (BoNT-A, rehab, AFO) can alter the natural history of spastic gait in CP and that improvements persist over 5 years. It is important to phrase the conclusion carefully to avoid implying a causal relationship without a control group. Suggest rewording to: “Our observational data suggest that integrated, repeated BoNT-A treatment plus intensive rehab is associated with long-term improvements…”
Summary of Discussion & Conclusions Review
Author Response
Thank you very much for taking the time to review this manuscript. Please find the detailed responses below and the corresponding corrections highlighted in the re-submitted files.
Comments 1: Title (Lines 1–4)
Consider clarifying that this is a retrospective observational study (per lines 11–12). For transparency, appending “... A 5-year Retrospective Observational Study” might increase clarity.
Response: Thank you for your comments. The title was changed accordingly
Comments 2: Abstract (Lines 6–27)
The abstract follows a structured approach (Introduction, Methods, Results, Conclusion), which is beneficial for clarity.
Comments 3: The conclusion highlights a significant, long-term positive effect. While that is valid, it would be good to explicitly recognize in the conclusion that this is observational data. Also emphasize that results may not be generalized to all clinical settings without caution.
Response: Thank you for your comments. The information was added.
Comments 4: Key Contribution (Lines 21–27)
The statement on lines 25–27 about “development of serious gait problems not observed” is quite strong. Clarify or temper this if the results only show that major deterioration was seen in only a small fraction, rather than “not observed” at all.
Response: Thank you for your comments. The information was added.
- Introduction (Lines 29–52)
Comments 5: Lines 32–34: The text states that between 75 and 91% of CP patients have spasticity interfering with motor function. The cited study discusses the prevalence of spasticity in CP (Are 75-91 correct?), it doesn't specifically address whether this spasticity "interferes with motor function.
Response: Data are not only from the Victorian registry but also from a systematic review enclosed in the article. For the second part, thank you for pointing out our mistake. It was part referring to Graham et al. [3]. Reference was added.
Role of Botulinum Toxin
Comments 6: Lines 35–40: The discussion of BoNT-A as an “important treatment modality” is well justified, highlighting its role in reducing hypertonia and preventing secondary complications. To further strengthen this section, the authors could also emphasize the growing research trends and increasing clinical adoption of BoNT-A for spasticity management (doi: 10.3390/toxins16040184).
The phrase “helping achieve maximal potential development” is somewhat broad;
Response: We added information with reference in the conclusions part.
Methods (Lines 53–124)
Comments 7: Lines 53–55: States that it is a single-center retrospective observational study at the Mazovian Neuropsychiatry Center in Poland. Clarify exactly how retrospective data were extracted and if any standardized protocol existed during the data collection. Inclusion Criteria Since the study focuses on gait, it is logical to exclude GMFCS V. This is appropriate, though it might be helpful to state why GMFCS V was excluded (non-ambulatory, cannot perform reliable gait assessments). Exclusion includes “other forms of CP, especially dyskinetic type,” GMFCS V, and orthopedic surgeries. You may also specify if children with prior rhizotomy or single-event multi-level surgeries (SEMLS) were excluded.
Response: Thank you for the correction I added this information in the section material and methods, settings and inclusion criteria.
Comments 8: Lines 66–74: A yearly frequency (1–2 times), with total doses ABOBoNT-A 20–30 U/kg or OnabotulinumtoxinA 10–20 U/kg. Typically multi-level injections.
Specify whether the brand was chosen randomly or systematically, and if the dosing conversions between abobotulinumtoxinA and onabotulinumtoxinA were consistent with known ratio guidelines.
State if sedation protocols or local anesthetics were standardized.
Mention the average interval between injections (3–6 months?). Currently, it says 1 to 2 times a year, but the typical recommended minimum interval is ~3 months.
Response: We added mean [SD] values. The same toxin brand was usually used for one patient, but random changes occurred during the observational period. We did not use strict dosing conversions. Mild sedation motioned in the text was standardized, and we added the drug name to the text. For over twenty years, we have tried to administer BoNT A as infrequently as possible, with an average of six months for the entire population. This approach seems rational in light of recent reports on possible muscle atrophy.
Comments 9: The rehabilitation intensity is described (3 to 12 weeks per year of “intensive” therapy, plus community-based therapy 2 to 5 times per week). However, the large variability in frequency (3–12 weeks) might introduce significant heterogeneity in outcomes. Consider discussing how you controlled or accounted for differences in therapy dose.
Response: Indeed, this may be a weakness of our study: lack of precise data to differentiate patients based on the therapy dose. However, all patients continued therapy at home and in day care units according to our guidelines, which may further obscure the differences revealed by such stratification.
3.3 Data Collection Procedure
Comments 10: Lines 91–103: Clarify how standardization of video capture was ensured (e.g., were children walking barefoot or with AFOs? Same walkway length? Same camera angles?). This is crucial for consistent OGS scoring. Indicate whether the OGS scorers were blinded to injection history or the timing of injections. Lines 95–96: Six gait assessments are selected for each patient over 5 years, including pre- or at-least-3-months post-BoNT-A.This is logical, but are these time points standardized (e.g., 0, 12, 24, 36, 48, 60 months) or did it vary by patient? In retrospective studies, timing might differ widely if the child missed an appointment or received injections off schedule.
Response: Thank you for the correction I added this information in the section material and methods, data collection procedure. The OGS assessor was not blinded for BoNT injection timing, but due to the retrospective nature of the study, even exposition to the toxin in the study group and the required time period from injection, seems to be not significant.
3.4 Statistical Analysis
Comments 11: Line 115: The text says “detailed scores for the MSt foot and knee position... have been modified to eliminate duplicate values.”
This might raise concerns. If you collapsed categories or changed scoring, it can alter validity. Clarify precisely which values were merged or recoded and why that does not affect the total OGS scoring.
Indicate how missing data (if any) were handled. Were any participants dropped from analysis if they missed one of the 6 visits?
Response: Thank you for the correction I added this information in the section material and methods, statistical analysis
Results (Lines 125–164)
Comments 12: Lines 125–128: You report 120 males, 80 females, with ages 24–46 months (mean 32.23 ± 6.96). GMFCS distribution is clearly stated: I (3%), II (48%), III (36%), IV (13%). Consider adding the exact n for each GMFCS level (e.g., n=6 for level I, n=96 for level II, etc.).
Response: Thank you for the comment. Values were added.
Comments 13: OGS Total Score
Lines 129–132: States total OGS scores “grew” over the period (p<0.001), with significant difference between the 1st and 6th assessments. Also indicates 74.5% and 76.5% improvement in right and left limbs, respectively, while 8.5% and 7% deteriorated. Adding median (or mean) OGS changes would further help quantify how large the improvement was. The tables do show median changes, but summarizing that in the text might be clearer.
Lines 133–164: You systematically describe the results for foot contact at IC, foot contact at midstance, timing of heel rise, knee position at midstance and terminal stance, hip position at terminal stance, etc. This thorough reporting is good; however, the text is quite dense. For clarity, you might reorganize so that each joint is summarized in a single paragraph (foot → knee → hip).
Mention some absolute numbers or percentages for each domain, plus the direction of change, which is partly done. For instance, lines 139–142 mention that toe-walking at midstance decreased from 64% to ~39%.
As it is a retrospective study, mention if any data points were missing. Did all 200 children complete all 6 assessments? The text implies yes, but please confirm.
Response: Thank you for the correction I added some information in the section results and reorganize results that each joint is summarized in a single paragraph. We try not duplicate tabels in te text.
- Discussion & Conclusions (Lines 165–281)
Comments 14: Lines 250–260: The paper indicates that the improvements were stable or increased across the 5-year span. This addresses a common question about whether repeated BoNT-A leads to diminishing returns. Cite guidelines or systematic reviews that address repeated BoNT-A injections in CP.
Response: Thank you for your comments. We already cited it in the introduction part.
Strengths and Limitations
Comments 15: Lines 261–269: Strengths include large sample size and standardized therapy center. Limitations include retrospective design, no control group, no 3D gait analysis.
The discussion of limitations is appropriate, but consider discussing how lack of randomization or standardized home therapy may have introduced confounding. Also mention the wide variance in physiotherapy “intensity” (3–12 weeks).
Response: Thank you for your comments. The information was added.
Comments 16: Lines 278–281: Concludes that integrated treatment (BoNT-A, rehab, AFO) can alter the natural history of spastic gait in CP and that improvements persist over 5 years. It is important to phrase the conclusion carefully to avoid implying a causal relationship without a control group. Suggest rewording to: “Our observational data suggest that integrated, repeated BoNT-A treatment plus intensive rehab is associated with long-term improvements…”
Response: Thank you for your comments. I did rephrase it accordingly.

Round 2
Reviewer 1 Report
Comments and Suggestions for Authors
Thr authors have appropriately modified the manuscript and have provided
comprehensive responses to all the questions and observations raised. I would only further
suggest discussing the responses within the text, for example, by explaining why it was not
possible to include a control group, or why a post-infiltration control to study the direct denervation effect of the toxin was unnecessary. Additionally, it would be helpful to specify that the patients did not receive oral therapies and that no treatment-related serious adverse events (SAEs) were encountered.
Author Response
Thank you very much for taking the time to review this manuscript again. Below are the detailed responses along with the corresponding corrections highlighted in the resubmitted files.
Comments: The authors have appropriately modified the manuscript and have provided
comprehensive responses to all the questions and observations raised. I would only further
suggest discussing the responses within the text, for example, by explaining why it was not
possible to include a control group, or why a post-infiltration control to study the direct denervation effect of the toxin was unnecessary. Additionally, it would be helpful to specify that the patients did not receive oral therapies and that no treatment-related serious adverse events (SAEs) were encountered.
Response: Thank you for your feedback. We did add information about oral antispastic medication and treatment-related serious adverse events (SAEs) in the Results and Materials and Methods sections, respectively.

Reviewer 2 Report
Comments and Suggestions for Authors
The manuscript has been improved after revisions. Thank you for point by point responses.
Author Response
Thank you very much indeed for taking the time to review this manuscript again.
